# Morbidity and Mortality of Neutropenic Patients in Visceral Surgery: A Narrative Review

**DOI:** 10.3390/cells11203314

**Published:** 2022-10-21

**Authors:** Ann-Kathrin Lederer, Fabian Bartsch, Markus Moehler, Peter Gaßmann, Hauke Lang

**Affiliations:** 1Department of General, Visceral and Transplant Surgery, University Medical Center of the Johannes Gutenberg University, 55131 Mainz, Germany; 2Center for Complementary Medicine, Department of Medicine II, Medical Center-University of Freiburg, Faculty of Medicine, University of Freiburg, 79106 Freiburg, Germany; 3First Medical Clinic and Policlinic, University Medical Center of the Johannes Gutenberg University, 55131 Mainz, Germany

**Keywords:** leukocyte disorders, outcome, leukopenia, agranulocytosis, hematologic diseases, postoperative complications, surgery

## Abstract

Leukocytes are essential for the function of the immune system and cell–cell interaction in the human body, but hematological diseases as well as chemotherapeutic treatments due to cancer lead to occasionally or even permanent leukocyte deficiency. Normally, more than 50% of leukocytes are neutrophilic granulocytes, and leukopenia is, therefore, mostly characterized by a decrease in neutrophilic granulocytes. The consequence of neutropenia is increased susceptibility to infection, but also healing disorders are suggestable due to the disturbed cell–cell interaction. While there is no surgical treatment for leucocyte disorders, patients suffering from neutropenia are sometimes in need of surgery for other reasons. Less is known about the morbidity and mortality of this patients, which is why this narrative review critically summarizes the results of recent research in this particular field. The results of this review suggest that neutropenic patients in need of emergency surgery have a higher mortality risk compared to non-neutropenic patients. In contrast, in elective surgery, there was not a clear tendency for a higher mortality risk of neutropenic patients. The role of neutrophilic granulocytes in inflammation and immunity in surgical patients is emphasized by the results, but most of the evaluated studies showed methodological flaws due to small sample sizes or risk of bias. Further research has to evaluate the risk for postoperative complications, particularly of infectious complications such as surgical site infections, in neutropenic patients undergoing elective surgery, and should address the role of neutrophilic function in postoperative morbidity and mortality.

## 1. Introduction

Disorders of the cellular component of the human immune system may lead to severe disturbances in the homeostasis as body functions are regulated and affected by a variety of cytokines, which are secreted by immune cells such as leukocytes [1,2,3,4]. A decrease in the concentration of leukocytes below 4000 cells per µL blood is called leukopenia [5]. Leukocytes are the first responders of innate immunity, and their functional alterability due to internal and external signals is remarkable [2,6]. In case of infection, leukocytes are rapidly mobilized to sites of infection [7,8]. The lifespan of blood-circulating leukocytes and tissue leukocytes is approximately 24 h dying due to apoptosis after the completion of antimicrobial function and being eliminated by macrophages [2,6,7,9].

Congenital leukocyte disorders are rare, while acquired disorders are a commonly encountered disease pattern in Western countries [5,10]. Destruction of the bone marrow, commonly seen in hematological diseases, might lead to leukopenia as the production of leukocytes is located in the bone marrow. Further reasons for leukopenia include vitamin deficiency (especially vitamin B_12_), folic acid and copper deficiency, toxic and radiation damage, as well as autoimmune diseases and infectious diseases (such as HIV and SARS-CoV-2) [10,11,12]. Several drugs are known to be associated with leukopenia by inducing apoptosis, inhibiting leukocyte development, or being toxic for myeloid precursor cells [13]. In Western countries, one of the most common reasons for leukopenia is chemotherapeutic treatment due to cancer.

Usually, more than 50% of leukocytes are neutrophilic granulocytes. Therefore, leukopenia is mostly characterized by a decrease in neutrophilic granulocytes, which is why leukopenia and neutropenia are often used interchangeably [2]. Palmblad et al. defined three types of neutropenia from mild to severe (Table 1) [14]. This grading implies also a clinical consequence as severe neutropenia is associated with an increased risk of infection, especially with opportunistic pathogens, that are not harmful in immunocompetent patients [14,15]. The decrease in neutrophilic granulocytes is one of the main contributors for the occurrence of infection in cancer patients receiving chemotherapy or suffering from hematological diseases [16,17].

While there is no surgical treatment for leucocyte disorders, patients suffering from leukopenia are sometimes in need of surgery for other reasons. Independent of their hematological diseases, some of the patients develop appendicitis or acute cholecystitis, for example, indicating the necessity for surgery [19]. Clinically, the symptom presentation of these patients is often atypical, leading to misdiagnosis and deteriorating morbidity and mortality [19,20]. Moreover, the general condition of most of these patients is significantly compromised due to their primary disease, thus emphasizing the necessity to realistically estimate the risk of surgery. Low-risk routine interventions in immunocompetent patients might be life-threatening for leukopenic/neutropenic patients. The surgical risk of these immunocompromised patients might exceed the potential benefits and non-surgical treatment may be superior. To date, little is known about surgical risks of patients with preoperative leukopenia/neutropenia, and there is hardly any evidence for reasonable recommendations concerning surgery in the particular situation of leucopenia. This narrative review puts a special focus on patients suffering from preoperative neutropenia, in the need for abdominal surgery, and sought to summarize recent studies evaluating patients’ morbidity and mortality to give a realistic overview of potential treatment approaches in this extremely vulnerable cohort.

## 2. Materials and Methods

We performed a narrative literature review searching databases of Medline and Web of Science from inception to August 2022. All kind of human studies, except for case reports and case series, were eligible for evaluation. All adult patients regardless of gender and all general surgical and visceral surgical procedures were considered. Eligible studies had to report on preoperative leukopenia/neutropenia and outcomes after visceral surgery. Due to the often interchangeable usage of leukopenia and neutropenia, we decided to use both terms for searching. Leukopenia was defined as a white blood cell count lower than 4000/µL, and the definition of neutropenia is shown in Table 1.

Search terms were “preoperative AND (leukopenia OR neutropenia) AND surgery” as well as “emergency surgery AND (leukopenia OR neutropenia)”. To contextualize the results of included studies, the discussion was completed by a selective literature search. Abstracts were screened without a language barrier, and a full-text evaluation was only performed in English, French, and German. No further search restrictions were applied. The manuscript was prepared according to recent recommendations for the preparation of narrative reviews, as well as to the Scale for the Assessment of Narrative Review Articles (SANRA) to ensure quality [21,22].

### Research Questions

The following research questions were posed before starting a database research:(1)Does preoperative neutropenia affect the morbidity and mortality of patients undergoing emergency visceral surgery?(2)Does preoperative neutropenia affect the morbidity and mortality of patients undergoing elective visceral surgery?

## 3. Results

Overall, 1147 results were evaluated and 17 reports were included for final analysis. The process of screening and selection is shown in Figure 1. All of the included trials except of one pilot trial were retrospective cohort analyses. An overview of all included studies is provided in Table 2.

### 3.1. Research Question 1: Morbidity and Mortality in Emergency Surgery

The first research question aimed to clarify whether preoperative neutropenia affects the morbidity and mortality of patients, who are in need of emergency visceral surgery. Overall, eight studies including more than 3500 neutropenic/leukopenic patients were evaluated (see Table 1). All data, except for one pilot trial [23], were captured retrospectively. Most of the trials reported an association between patients’ mortality and a preoperative leukopenia/neutropenia, but were limited in parts due to an incomplete reporting of detailed surgical or patient-related data, as well as small sample sizes and methodological shortcomings, as discussed hereinafter.

Sullivan et al. compared data of 956 chemotherapy (<30 days prior to surgery) patients who underwent emergency surgery to 956 non-chemotherapy patients who underwent a similar emergent procedure [25]. Detailed information about the indication for surgery was not reported, but most of the patients received partial colon resection (17%), followed by a partial resection of the small intestine (11%). Chemotherapy patients had significantly lower levels of leukocytes and a greater frequency of leukopenia preoperatively. Leukopenia was found to be an independent predictor of death. The mortality of chemotherapy patients was significantly higher than the mortality of non-chemotherapy patients (22% vs. 10%, *p* < 0.001). Furthermore, postoperative complications, particularly major complications, occurred significantly more frequently in chemotherapy patients (44% vs. 39%, *p* = 0.033). The study is limited by the comparability of groups as non-chemotherapy groups did not suffer from cancer, but significantly more often from cardiac diseases. For decades, it is discussed that cancer is a systemic disease, which is why cancer patients may not be comparable to non-cancer patients [39,40]. The authors considered this as they performed a subgroup analysis excluding patients with disseminated malignant disease. Mortality remained significantly higher in the chemotherapeutic group (17% vs. 10%, *p* = 0.002). Nevertheless, general preoperative characteristics, such as BMI, weight loss, diabetes, alcohol consumption, and steroid use, differed significantly between chemotherapy and non-chemotherapy patients [25]. Moreover, cachexia, which often occurs in cancer patients, is an independent risk factor for worsening survival [41]. Furthermore, research results suggest that suffering from diabetes and alcohol abuse might affect postoperative complication rates [42,43]. Steroid use is a well-known risk factor for postoperative infectious complications [44]. The results in the trial by Sullivan et al. might therefore be influenced by factors other than leukopenia as well.

Sudarshan et al. reported that preoperative leukopenia has been revealed as an independent predictor of death in an inhomogeneous cohort of 527 patients who underwent emergency general surgery (both minor and major). It cannot be ruled out that neutropenia is a consequence of immunosuppression due to a severe sepsis, which is a well-known factor for poor outcome [26,45]. The thesis reporting that preoperative sepsis-induced leukopenia is associated with an increased mortality is supported by further publications by Hansen et al., who reported 105 patients who underwent emergency surgery due to diverticulitis [24], and Mokart et al., who evaluated the outcome of 58 hematology patients (some at just a few weeks after stem cell transplantation, but only 17 leukopenic patients) who underwent emergency surgery mostly due to abdominal pain [28]. A recently published analysis by Joo et al. aimed to assess the outcome of patients with left colonic perforation, not only due to perforated diverticulitis, but also due to perforated cancer or iatrogenic genesis. The authors stated that preoperative leukopenia was an independent risk factor for mortality, but the sample size of leukopenic patients (*n* = 12) was very small [29].

A very large database analysis by Gulack et al. evaluated 2057 leukopenic patients who underwent visceral emergency surgery, revealing a significantly higher morbidity of leukopenic patients compared with non-leukopenic patients (45% vs. 27%). Patients’ demographics differed significantly between leukopenic and non-leukopenic patients (BMI, ASA class, presence of ascites, weight loss, renal failure, steroid use, as well as previous radiation or chemotherapy), and leukopenic patients suffered significantly more frequently from sepsis (20% vs. 10%) and septic shock (28% vs. 8%); however, after adjustment for patient-related factors, leukopenia remained as a significant predictor for mortality. The results are strengthened by the large sample size and the well-planned analysis. The authors decided to exclude patients over the age of 89 as well as patients with disseminated cancer.

In addition to the above-mentioned supposed higher mortality rate of neutropenic patients, the included studies indicate that neutropenic patients are threatened by diseases which are not typical in immunocompetent patients. For example, Wade et al. reported results of 50 mostly hematological neutropenic patients suffering from acute abdominal pain [20]. Besides typical surgical emergencies, such as cholecystitis, intestinal perforation, or intra-abdominal abscesses, patients suffered from invasive infection due to *Aspergillus fumigatus* or from neutropenic enterocolitis. Surgery was performed in 17 patients, and 41% (*n* = 7) died after surgery (four due to multiple organ failure and three due to invasive infection with *Aspergillus fumigatus* proven by autopsy). The authors emphasize the incongruity of clinical symptoms, suspected diagnosis, and intraoperative findings in neutropenic patients, and that rare causes for abdominal pain are expected. The invasive infection with *Aspergillus fumigatus* is associated with a high mortality rate up to 50% in immunocompromised patients, despite antifungal treatment [46]. Recent research indicates that the mortality rate of a neutropenic enterocolitis (also known as typhlitis) is up to 50%, possibly leading to an intestinal perforation and subsequent sepsis [47,48]. Every neutropenic patient developing fever and abdominal pain is suspected to suffer from neutropenic enterocolitis. Further symptoms may include abdominal distension, nausea/vomiting, and diarrhea, but it is also possible that patients show no specific symptoms [47]. A neutropenic enterocolitis occurs 10–14 days after the initiation of chemotherapeutic treatment, and is located in the cecum and sometimes in the terminal ileum [49]. CT imaging is helpful for diagnosis and shows typically bowel wall thickening, a dilated cecum, local inflammatory signs in the right lower quadrant, as well as pericecal fluid [50,51]. A neutropenic enterocolitis is often misdiagnosed, and the pathogenesis of neutropenic enterocolitis is barely understood [20]. Histological examinations reveal mucosal or whole-wall edema, hemorrhage, ulceration, and mucosal or transmural necrosis to be signs of a necrotizing inflammation [52,53]. The recommended treatment of neutropenic enterocolitis is non-surgical, but in case of intestinal perforation, urgent surgery is indicated to avoid any development of abdominal sepsis [53].

Right-sided abdominal pain and diarrhea, particularly bloody diarrhea, might also be caused by non-occlusive mesenteric ischemia (NOMI), which typically occurs in patients with circulatory disorders, but is also reported in neutropenic patients [54]. For example, Tanaka et al. reported about a 74-year-old man diagnosed with oropharyngeal carcinoma who developed febrile neutropenia followed by septic shock due to NOMI on day 9 of chemotherapeutic treatment [55]. The occurrence of NOMI is a life-threatening event affecting particularly critically ill patients and showing a mortality rate up to 60% [54]. In general, surgical treatment is not the preferential approach in NOMI as the pathogenesis is a mesenteric vasospasm, which cannot be cured by surgery [56]. Fluid resuscitation, cardiac output optimization, and vasopressor elimination are recommended in the guidelines of the World Society of Emergency Surgery [57]. Nevertheless, in case of progressive ischemia and development of necrotic bowel, surgery is the method of choice to control abdominal infection and subsequent sepsis.

In summary, an evaluation of publications revealed interesting results on the postoperative outcome of neutropenic patients; however, as almost every publication is of retrospective nature and some studies are more than 10 years old, the results become influenced by several confounding factors. Recent research data uncover the difficulty in comparing neutropenic patients with immunocompetent control groups as leukopenia/neutropenia appears to be often associated with other critical conditions, such as sepsis or organ failure. Furthermore, patients’ co-existing diseases and medications such as steroid treatment make it difficult to evaluate the sole impact of neutropenia on postoperative mortality and morbidity. Nevertheless, the results emphasize the necessity of being highly attentive as symptoms and disease patterns of neutropenic patients might be different to immunocompetent patients. If in doubt, a surgical colleague should be consulted to evaluate abdominal pain in a neutropenic patient, and rapid radiologic diagnostic should be performed. Even if the mortality rate of neutropenic patients might be higher compared to non-neutropenic patients, surgery might be the only approach for survival in case of impending abdominal sepsis. Similar to other critically ill patients, it appears that prompt diagnosis followed by an early and specific treatment is essential for the outcome of neutropenic patients.

#### Short Digression: Improving the Postoperative Outcome by an Application of G-CSF

With regards to assuming a potential role of neutrophilic granulocytes in the development of postoperative complications, it is also assumable that preoperatively applying the granulocyte colony-stimulating factor (G-CSF) might be helpful to prevent postoperative complications in neutropenic patients. Nishida et al. reported the results of a small pilot trial, which evaluated the impact of a preoperative application of granulocyte colony-stimulating factor (G-CSF) in neutropenic patients, who were in need of emergency surgery due to gastrointestinal perforation [23]. Eight patients received G-CSF subcutaneously (150 microgram/day) during the perioperative period, and were compared to twenty-three patients who did not receive G-CSF. Only 1 G-CSF patient (13%) died, but 15 patients (65%) in the non-G-CSF group died due to abdominal sepsis. Meanwhile, the results are more than 30 years old, and limited due to the small sample size. The idea of G-CSF application to prevent infectious postoperative complications due to a well-known immunodysfunction after surgery was later evaluated by several trials. For example, Schneider et al. conducted a randomized controlled trial on non-neutropenic patients who underwent major abdominal surgery [58]. Sixty patients received either placebo (twenty patients) or G-CSF (forty patients, 15 μg/kg body weight) over six perioperative days, either as three bolus administrations of 5 μg/kg body weight or as a continuous administration of 2 μg/kg body weight following an initial bolus of 5 μg/kg body weight. Interestingly, 30% of non-G-CSF patients and 13% of G-CSF patients developed infectious complications, respectively.

Another randomized controlled trial by Schaefer et al., who treated 77 non-neutropenic esophageal cancer patients with G-CSF (daily dose of 300 μg in patients below 75 kg body weight and 480 μg if body weight exceeded 75 kg, starting two days prior to surgery and terminating on the seventh postoperative day) and compared the results to 76 non-neutropenic esophageal cancer patients without G-CSF treatment, did not reveal any difference regarding their infectious complication rate (43% in the placebo group vs. 44% in the G-CSF group) [59]. Nevertheless, both studies included non-neutropenic patients, which is why no recommendation regarding G-CSF application can be made for neutropenic patients. It appears that G-CSF application might be helpful in neutropenic patients to prevent infectious complications, but data on this are widely lacking. A case report by Nakano et al. illustrated the role of G-CSF to prevent postoperative infectious complications in a chronic neutropenic patient suffering from acute cholecystitis, but also emphasized that a critical evaluation is crucial to prevent potential adverse events [60].

### 3.2. Research Question 2: Morbidity and Mortality in Elective Surgery

The second research question aimed to clarify whether preoperative neutropenia affects morbidity and mortality in elective visceral surgery. Nine studies on more than 3300 neutropenic/leukopenic patients were evaluated (see Table 1). All data were captured retrospectively. Studies were limited mostly due to small sample sizes or methodological shortcomings, as well as a missing differentiation between emergency and elective surgery. Many of the included studies showed significant differences between the morbidity and mortality of neutropenic patients compared to non-neutropenic patients, but further analysis controlling for confounding factors did not confirm the results.

Overall, the addressing of this question by literature review was challenging as elective surgery is usually obviated in neutropenic patients, whenever possible. Nevertheless, today, it is well established that surgery is promptly feasible in patients after chemotherapy. It is known that chemotherapy leads to depression in the hematopoietic system, but it is mostly short-lasting and restores in less than six weeks. In clinical surgery, for example, an interval of six weeks between radio-chemotherapy and surgery for rectal cancer is recommended. This is supported, for example, by Fokstuen et al. who compared 144 surgery-alone rectal cancer patients with 130 patients who had short-course radiation prior to rectal surgery less than 10 days after radiation [30]. Irradiated patients had significantly lower levels of leukocytes compared to non-irradiated patients, and 31% of irradiated patients had leukopenia prior to surgery. Interestingly, irradiated patients showed a significantly less leukocyte response after surgery (measured by a pre- to postoperative leukocyte ratio), which was associated with the risk of developing sepsis. The postoperative morbidity of irradiated patients differed significantly among non-irradiated patients (47% vs. 25%, *p* < 0.001), mostly due to infectious complications. In another study by Grant et al., data on 4369 patients who received chemotherapy for malignancy less than 30 days prior to elective or emergent abdominal surgery were evaluated [36]. Elective surgery was performed in 63% of patients, and 20% of patients had preoperative leukopenia. The morbidity and mortality of leukopenic and non-leukopenic patients differed significantly, but no statistical significance was revealed after controlling for confounding factors. The authors did not differentiate between elective and emergency surgery, which is a clear limitation of the study. Another trial which did not differentiate between elective and emergency surgery was reported by Pluta et al., who tested the serum concentration of leukocytes in 101 patients who underwent major gastrointestinal surgery [34]. They found no significant difference in leukocyte counts in survivors and deceased, and no prediction regarding the patients’ outcome. Interestingly, Reim et al. postulated that clinical inapparent leucopenia after neoadjuvant chemotherapy does not affect the postoperative complication rate in elective gastric cancer surgery [31]. The authors analyzed data of 214 patients and found that patients with a preoperative leukopenia had slightly more postoperative complications than patients with a normal leukocyte count preoperatively, but deeper analysis did not reveal any statistical significance. Similar results were found by Ohira et al., who reported on the postoperative complication rate of esophageal cancer patients [33], and by Hara et al., who also evaluated data of esophageal cancer patients comparing 52 patients with grade 2 to 4 leukopenia to 100 patients with grade 0 to 1 leukopenia [37]. Chen et al. postulated that a severe neutropenia might be able to predict postoperative major complications after liver resection in colorectal cancer patients with hepatic metastases [35]. In their cohort analysis of 141 patients, 50% of patients developed postoperative complications including 28 major complications, whereas only 9 were stated to be related to surgery, but the authors did not report any complications in detail, making the results unconvincing.

Focusing on the evaluation of typical postoperative complications, such as surgical site infections (SSIs), it appears that data of neutropenic patients are even more sparse. In general, approximately 25% of visceral surgical patients develop SSI, making it one of the most frequent hospital-acquired infections [61,62,63]. According to Horan et al., SSIs are divided into superficial SSI (skin or superficial tissue), deep incisional SSI (deep soft tissue such as fascial and muscle layers), and organ/space SSI, which occur within 30 days after surgery [63,64]. The origin of SSI is complex, but a disequilibrium of microbial commensals which destructs the inner barrier against potentially pathogenic bacteria-promoting bacterial translocation is suggested [65]. It is plausible that immunological disturbances might be able to aggravate the pathogenetic process of SSI by opening the door for microbes, which are harmful for immunocompromised patients. Zarain-Obrador et al. evaluated data of 1727 patients, who underwent colorectal surgery given their risk of SSI [38]. Sixteen percent of patients had emergency surgery. The authors compared patients who received a surgical care bundle (involved replacing chlorhexidine handwashing with hydroalcholic handwashing, replacing razor blade shaving with electric shaving, antisepticising the surgical field with alcoholic chlorhexidine, and assessing surgical antibiotic prophylaxis) to prevent postoperative infections on patients without prophylaxis. Interestingly, they found no significant differences between the groups, but neutropenia was revealed as an independent risk factor for developing SSI. However, the authors did not report on patients’ pre-existing illnesses, which is known to contribute to the development of SSI [66]. Bamba et al. evaluated the risk of infection within 30 days after port implantation, and found that low preoperative leukocytes were a risk factor for an early infection [32]. The results are limited due to the small sample size, but emphasize the necessity of being aware of early port infection in neutropenic patients.

As a conclusion, it is not suggested by the results that neutropenic patients are at high risk for postoperative complications or even postoperative death. There might be a higher risk for infectious complications, but data which report on infectious complications in detail are widely lacking. An indication for elective surgery in neutropenic patients should still be made carefully. Ideally, future clinical trials which evaluate the efficacy of neoadjuvant therapies should also report on the normalization of neutrophilic granulocytes immediately before surgery to clarify the role of chemotherapy-induced neutropenia in the development of postoperative complications in elective cancer surgery.

## 4. Conclusions

The results emphasize the role of neutrophilic granulocytes in inflammation and immunity in surgical patients, but it appears to be extremely difficult to assess the effect of neutropenia on postoperative morbidity and mortality of visceral surgery patients. The interactions between immune cells and a variety of other potentially impactful factors make it almost impossible to draw reliable conclusions. Furthermore, most of the evaluated studies showed methodological flaws due to small sample sizes or bias risks. Nevertheless, the results are interesting and emphasize that physicians have to be reckoned with atypical clinical patterns obscuring a potential severe life-threatening impact that requires urgent therapy. The results of this review suggest that neutropenic patients in need of emergency surgery have a higher mortality risk compared to non-neutropenic patients. In contrast, in elective surgery, there was not a clear tendency for a higher mortality risk of neutropenic patients, but future research should evaluate the risk for postoperative complications, particularly of infectious complications such as surgical site infections. Additionally, experimental research should help to clarify the role of neutrophilic function in postoperative morbidity and mortality.


## Figures and Tables

**Figure 1 cells-11-03314-f001:**
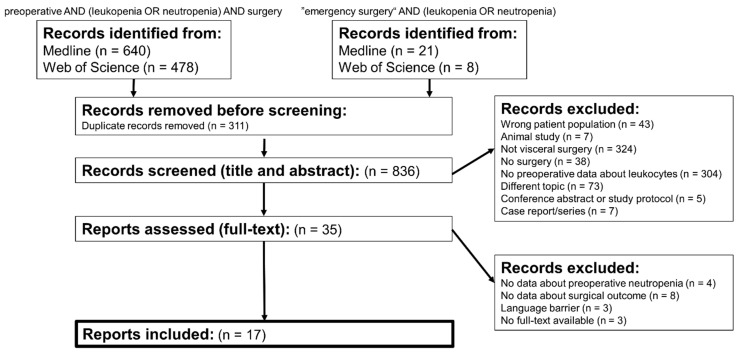
Flowchart of search process.

**Table 1 cells-11-03314-t001:** Types of neutropenia, modified according to Palmblad et al., the National Institutes of Health, and the National Cancer Institute [14,18]. * The grade of chemotherapy-induced neutropenia is not only evaluated by the concentration of neutrophilic granulocytes, but of all immune cells.

Type	Cells Per µL Blood	Chemotherapy-Induced *
	>1500	Grade I
Mild	1000–1500	Grade II
Moderate	500–1000	Grade III
Severe	<500	Grade IV

**Table 2 cells-11-03314-t002:** Overview of included studies.

Author	Ref.	Year	Study Type	*N* *	Type of Surgery	Disease	Results °
Wade et al.	[20]	1990	Retrospective	17	Emergency	Hematologic	Mortality 41%
Nishida et al.	[23]	1996	Pilot trial	8/23	Emergency	Perforation	Mortality 13% vs. 65% ^+^
Hansen et al.	[24]	1998	Retrospective	105	Emergency	Diverticulitis	Predictor for death
Sullivan et al.	[25]	2012	Retrospective	956/956	Emergency	Diverse	Predictor for death
Sudarshan et al.	[26]	2015	Retrospective	527	Emergency	Diverse	Predictor for death and complications
Gulack et al.	[27]	2015	Retrospective	2057/18386	Emergency	Diverse	Predictor for death
Mokart et al.	[28]	2017	Retrospective	17/58	Emergency	Hematologic	No association
Joo et al.	[29]	2020	Retrospective	12/79	Emergency	Colonic perforation	Predictor for death
Fokstuen et al.	[30]	2009	Retrospective	274	Elective	Rectal cancer	No association
Reim et al.	[31]	2010	Retrospective	58/156	Elective	Gastric cancer	No association
Bamba et al.	[32]	2014	Retrospective	33/66	Elective	Diverse	Association to an early port infection
Ohira et al.	[33]	2015	Retrospective	44	Elective	Esophageal cancer	No association
Pluta et al.	[34]	2018	Retrospective	101	Elective	Diverse	No association
Chen et al.	[35]	2019	Retrospective	141	Elective	Liver metastases	Predictor for major complications
Grant et al.	[36]	2020	Retrospective	891/3493	Elective	Malignant disease	No association
Hara et al.	[37]	2021	Retrospective	52/100	Elective	Esophageal cancer	No association
Zarain-Obrador et al.	[38]	2021	Retrospective	1727	Elective	Colorectalsurgery	Risk factor for surgical site infection

* In the case of a controlled study, *n* is reported as follows: *n*—study group and *n*—control group. ° Outcome of leukopenic/neutropenic patients or impact of preoperative leukopenia/neutropenia on morbidity and mortality. ^+^ All patients were neutropenic, but study group received G-CSF perioperatively; ref = Reference, ICU = intensive care unit.

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
