# Peer review of "Morbidity and Mortality of Neutropenic Patients in Visceral Surgery: A Narrative Review"

_cells, 2022, doi:10.3390/cells11203314_

Round 1

Reviewer 1 Report

The authors of “Morbidity and Mortality of Neutropenic Patients in Visceral Surgery – A Narrative Review” wrote an exemplary manuscript. The topic is very important, since to date little is known about surgical risks of neutropenic patients and it is necessary to sensitive the medical doctors to this issue.
Based on the recent literature the authors wanted to analyze, if neutropenia has an impact on morbidity and mortality in visceral surgery. The authors posed 2 specific questions:
I) Does preoperative neutropenia affect the morbidity and mortality of patients under-going emergency visceral surgery?
II) Does preoperative neutropenia affect the morbidity and mortality of patients undergoing elective visceral surgery?
Based on the summarized literature they suggest that neutropenic patients showed a higher mortality risk in emergency surgery. However, an effect of neutropenia, when patients undergo an elective visceral surgery, was not observed.
The authors have summarized and discussed the literature very well. The manuscript is quite interesting and very well written. I enjoyed reading the manuscript. Thus, I have no additional suggestion to improve the manuscript.

Author Response

Thank you for your review! We are very happy about your reply.

Reviewer 2 Report

This is a very well conceived and written review about risks in neutropenic patients undergoing visceral surgery. The research questions addressed are properly faced and comprehensive. Tables are clear (please, rename the second one as Table 2).  English language is fluent. 

Author Response

Thank you for your review! We renamed the second table to Table 2. Thank you for reading our manuscript so attentive!

Reviewer 3 Report

Lederer et al. submitted an interesting narrative review of the existing literature regarding the risk of mortality of neutropenic patients in emergency surgery vs. elective surgery. They thoroughly evaluated published studies regarding the effect of preoperative neutropenia on morbidity and mortality undergoing emergency vs. elective visceral surgery. They also shortly evaluate the possibility of improving the post-operative outcome by application of G-CSF.

This narrative review is concise and well-structured. The research questions addressed are valid and important to the field. The topic of the impact of neutropenia on morbidity/mortality of patients undergoing surgery is of high relevance and the gaps in knowledge regarding this topic are clearly identified in the review. The methodology used to address the research questions is well explained and easily understandable.

Comments:

1. Figure 1 should be added in higher quality, it looks a bit pixelated in the pdf document

2. English language should be edited. Some sentences are rather complex, long and difficult to understand.

Author Response

Thank you for your review!

We added a higher resolution version of Figure 1 to the manuscript.

We proofread the manuscript thoroughly, deleted unnecessary words and changed some phrases to make the manuscript more readable.